# Holographic Element-Based Effective Perspective Image Segmentation and Mosaicking Holographic Stereogram Printing

**Fan Fan** [1,2] **, Xiaoyu Jiang** [1,*]**, Xingpeng Yan** [1]**, Jun Wen** [1]**, Song Chen** [1]**, Teng Zhang** [1] **and Chao Han** [1]

[1] Department of Information Communication, Academy of Army Armored Forces, Beijing 100072, China; fanfan1912@163.com (F.F.); yanxp02@gmail.com (X.Y.); zqywenjun@gmail.com (J.W.); chensong_w@hotmail.com (S.C.); pofeite007@gmail.com (T.Z.); hansjohannes@163.com (C.H.)

[2] Institute of Construction and Development, Academy of Army Research, Beijing 100012, China

\* Correspondence: jiangxiaoyu2007@gmail.com

**Abstract:** Effective perspective image segmentation and mosaicking (EPISM) method is an effective holographic stereogram printing method, but a mosaic misplacement of reconstruction image occurred when focusing away from the reconstruction image plane. In this paper, a method known as holographic element-based effective perspective image segmentation and mosaicking is proposed. Holographic element (hogel) correspondence is used in EPISM method as pixel correspondence is used in direct-writing digital holography (DWDH) method to generate effective perspective images segments. The synthetic perspective image for holographic stereogram printing is obtained by mosaicking all the effective perspective images segments. Optical experiments verified that the holographic stereogram printed by the proposed method can provide high-quality reconstruction imagery and solve the mosaic misplacement inherent in the EPISM method.

**Keywords:** holographic printing; holographic stereogram; holographic element

---

## 1. Introduction

Holographic stereogram printing technology is used in many fields. Since Dennis Gabor invented holographic technology in 1948, holographic printing techniques can be categorized into three types: synthetic holographic stereogram printing, holographic fringe printing, and wavefront printing. The most widely used technology is holographic stereogram printing.

Holographic stereogram printing was first proposed by DeBitetto [1] in 1969. The perspective images sampled by a camera are exposed to a holographic recording medium by using a slit, generating a horizontal parallax holographic stereogram. In 1970, King [2] proposed a two-step horizontal parallax holographic stereogram printing technique. This two-step printing technique first records the master holographic plate, Then the second-step of the process is to record the reproduced image of the master holographic plate onto the transfer holographic plate. The production of a holographic stereogram that reproduces with white light and generates an orthoscopic real image is realized.

In 1990, Yamaguchi [3–5] proposed to print a full parallax holographic stereogram in a single step; based on this, a series of studies were carried out. In 1991, Halle [6–12] proposed a single-step holographic stereogram printing technique called Ultragram; this method processes the perspective image obtained by the sampling camera to generate a holographic stereogram that can be used for single-step printing. In this manner, arbitrary depth, full parallax, and undistortion holographic stereogram printing can be realized.

In 2001, Brotherton-Ratcliffe [13] proposed a technique for holographic stereogram printing by using pulsed lasers. In 2008, a single-step direct-writing digital holography printing technology [14,15] was proposed and used in Geola's holographic printing system. This technique employs a pixel corresponding method, replaces the image pixels loaded onto the spatial light modulator (SLM) with the pixels of the perspective image, and accurately reproduces the recorded scene. In 2017, Su and Yuan [16] proposed a method called effective perspective image segmentation and mosaicking (EPISM) method to simulate two-step method by a single-step process. Further research on the EPISM method was carried out by Su and Yan [17–19] to improve image quality and printing efficiency. In addition, many researchers have made valuable research in recent years [20,21].

The EPISM method uses the correspondence between the observation point and hogel to obtain an effective perspective images segments, and the resolution of the perspective image can be fully used by the EPISM method, but the use of an observation point will cause a mosaic misplacement of the reconstruction image when focusing away from the reconstruction image plane. To solve this problem, by using the idea of pixel correspondence of the DWDH method, the hogel correspondence is used as pixel correspondence to generate effective perspective images segments, and holographic element-based EPISM method is proposed.

This paper is divided into the following sections: In Section 2, the basic principles of the DWDH method, the EPISM method, and the holographic element-based EPISM method are introduced. In Section 3, the basic algorithms of holographic element-based EPISM method are given. In Section 4, the proposed method is verified by optical experiments, and the optical experiment results are compared with the experiment results of the EPISM method. In Section 5, conclusions are presented.

## 2. The Basic Principles

### 2.1. The Basic Principle of the DWDH Method and the EPISM Method

#### 2.1.1. The Basic Principle of the DWDH Method

The DWDH method converts sampled perspective images into a rearranged image for exposing. This algorithm is actually a pixel transformation from the film plane of the camera to the SLM plane of hogel. It is usually called 'I to S' transformation. The principle of the DWDH method is shown in Figure 1, There are six main planes: hologram plane, SLM plane, projected SLM plane, camera plane, film plane, and projected film plane.

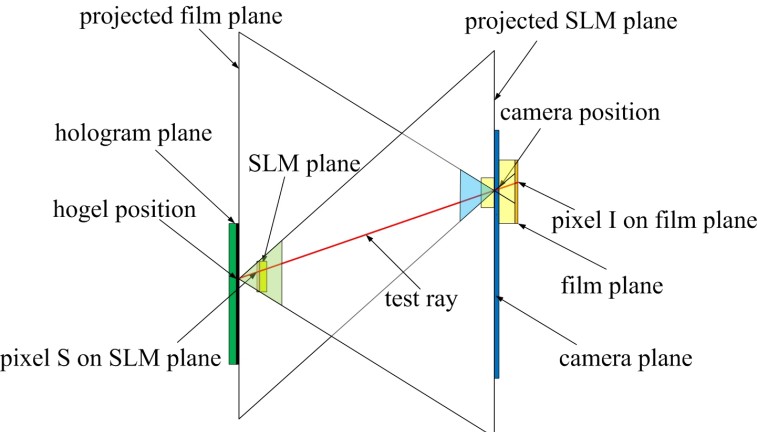

**Figure 1.** Ray-tracing principle of "I to S" transformation by DWDH method.

In this conversion, the camera lens and the print head of the holographic optical printer are regarded as a point, and a test ray passes through the camera lens and hogel. The pixel where the ray intersects the SLM plane is replaced by the pixel where the ray intersects the film plane, the 'I to S' transformation of a pixel is completed. All pixels on the SLM are replaced by pixels on the film plane,

a rearranged image for exposing is obtained. All the hogels on the hologram plane were exposed, and the DWDH holographic stereogram was obtained.

The pixel correspondence of the DWDH method can accurately reproduce the 3D scene, but the resolution of the sampled image is determined by the number of hogels; high-quality holographic stereograms often require hundreds of thousands of sampled images.

### 2.1.2. The Basic Principle of the EPISM Method

The EPISM method is proposed to achieve a two-step holographic stereogram printing effect by a single-step process. In the EPISM method, a liquid crystal display (LCD) panel is used as SLM. To achieve this purpose, by simulating the propagation process of information from different perspective images, the exposing synthetic perspective images for hogels on $H_2$ plate can be computer-generated directly. As shown in Figure 2, the EPISM method takes the center of the hogel on $H_2$ plate as an observation point. According to ray-tracing principle, when observing the hogel on virtual $H_1$ plate at point O, a viewing frustum with the observation point as the vertex and the hogel as the bottom is obtained. Since the reproduction image of the hogel on virtual $H_1$ plate is the corresponding perspective image, and the reproduction position is the position of the LCD panel when recording, the intersection of frustum and the reproduced image is the effective perspective image segments of the hogel for the observation point O. Mosaicking all effective perspective images segments together obtains a synthetic perspective image for recording onto the hogel of $H_2$. Record all the synthetic perspective images corresponding to the hogel on $H_2$ plate, and a full parallax holographic stereogram based on the EPISM method is received, as shown in Figure 2.

The resolution of the sampled image is determined by the resolution of the LCD panel in the EPISM method. Since the observation point is used on the hogel on the $H_2$ plate to generate effective perspective images segments, the influence of the hogel size on the effective perspective images segments is ignored, and this ignorance leads to slight mosaic misplacement in the position away from the reconstructed image plane, and the larger the dimension of the hogel on $H_2$ plate, the more obvious the mosaic misplacement.

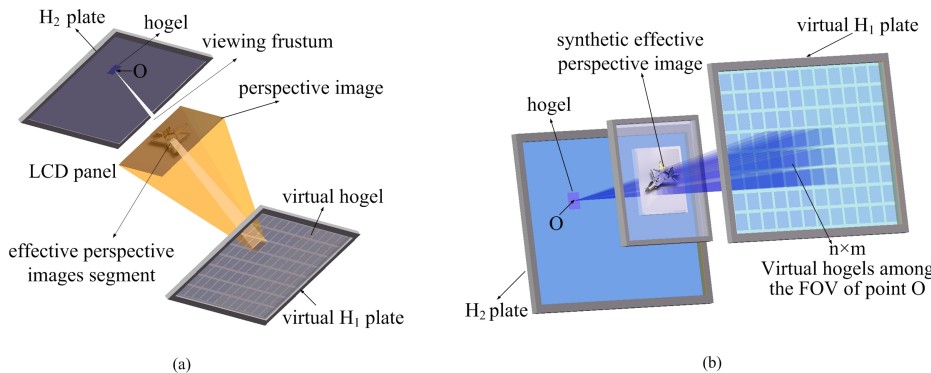

**Figure 2.** The primitive principle of the proposed method. (**a**) The extraction of effective perspective image segment corresponding to a single virtual hogel. (**b**) The synthetic effective perspective image mosaicked by effective images segments of multiple virtual hogels.

### 2.2. The Basic Principle of the Holographic Element-Based Effective Perspective Image Segmentation and Mosaicking (EPISM) Methods

Using holographic elements to replace pixels, the effective perspective images segments of adjacent hogels on virtual $H_1$ plate are mosaicked as shown in Figure 3, then the aliasing perspective images segments of adjacent hogel cannot be simulated by the LCD panel.

To avoid the aliasing of effective perspective images segments of adjacent hogels, the pixel correspondence in the DWDH method is used to establish the hogel correspondence. The DWDH method uses the hogel (as a point) position and the pixel coordinates on the SLM plane to determine

the camera position and the pixel coordinates on the film plane. As shown in Figure 4, if the dimension of hogel is considered, the pixel on the SLM plane in the DWDH method should be replaced by a segment of the LCD panel in holographic element-based EPISM method, and the dimension of the segment of the LCD panel is equal to the dimension of the hogel on virtual $H_1$ plate. The position of the hogel on virtual $H_1$ plate and the effective perspective images segments of its reconstruction images are determined by the position of the hogel on the $H_2$ plate and the segment of the LCD panel. A synthetic perspective image based on the holographic element can be obtained by mosaicking all the perspective images segments, and the aliasing of perspective images segments is avoided. Based on this idea, holographic element-based EPISM method is proposed.

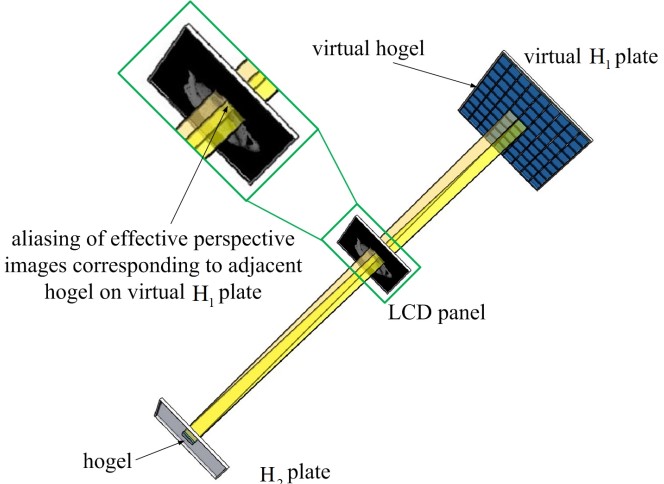

**Figure 3.** The effective perspective images segments of adjacent hogels are aliased When holographic element is used for the corresponding.

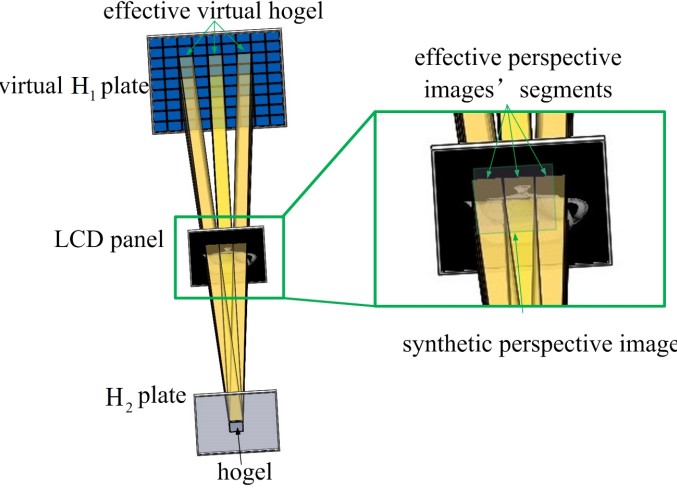

**Figure 4.** The holographic elements are used as pixels to solve the aliasing problem of adjacent hogels.

The principle of this method is described as follows: as shown in Figure 5, to treat a hogel as a pixel, this requires the same hogel dimensions on the virtual $H_1$ and $H_2$ plates, and the dimension of the effective perspective images segments are also the same as the hogel dimension, denoted as $l$. The dimension of the LCD panel is fixed, and the LCD panel is partitioned according to the dimension of the effective perspective images segments. It is necessary to choose a suitable distance to make the rectangle area formed by the hogel on $H_2$ plate and the segment of the LCD panel coincide with the hogel on the virtual $H_1$ plate. The reproducing image of the hogel on the virtual $H_1$ plate intersecting the segment of LCD generates the effective perspective image segments. For this purpose, as shown in

Figure 6, when the distance between the virtual $H_1$ plate and the LCD panel is denoted as $L_1$, we need to make $L_1$ be an integer multiple of $L_2$ which is the distance between the $H_2$ plate and the LCD panel. This ensures that the rectangular area formed by the hogel on $H_2$ plate and the segment of the LCD panel just falls on the hogel of virtual $H_1$ plate.

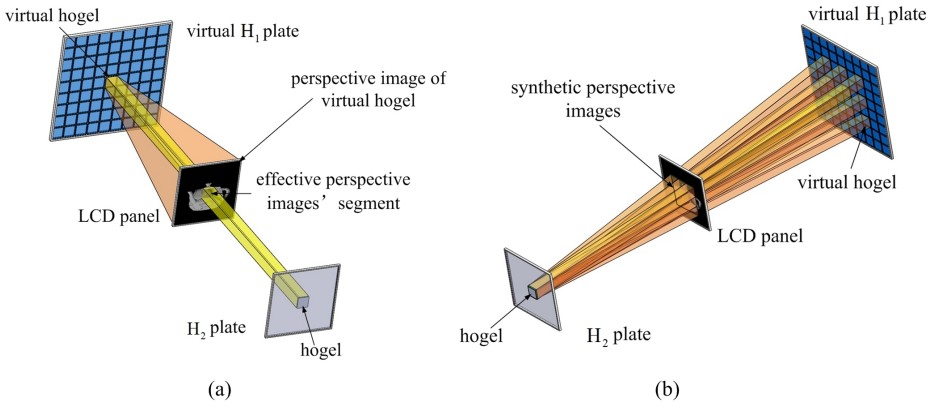

(a)                                    (b)

**Figure 5.** Holographic element-based effective perspective image segmentation and mosaicking methods (**a**) The extraction of effective perspective image segment corresponding to a single virtual hogel (**b**) The synthetic effective perspective image mosaicked by effective images segments of multiple virtual hogels.

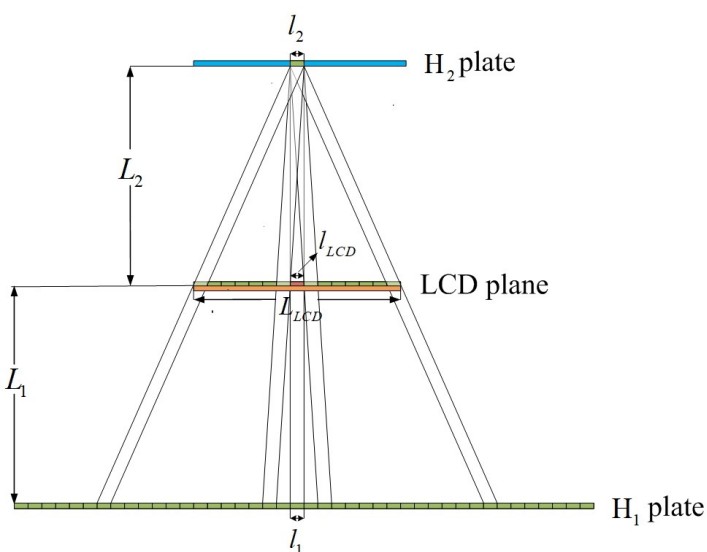

**Figure 6.** Parameter setting of holographic element-based EPISM methods.

## 3. The Basic Algorithm of the Holographic Element-Based EPISM Method

### 3.1. The Selection of the Hogel on Virtual $H_1$ Plate

First, hogel number of the $H_1$ plate should be fixed. As shown in Figure 6, to make full use of the LCD resolution, choosing the appropriate field of view (FOV) of the hogel on the $H_2$ plate and $L_2$ makes the display area of the LCD completely contained by the FOV. The display area dimensions of the LCD panel is denoted as $L_{LCD}$. Using $l_1$ and $l_2$ to represent the dimension of hogel on $H_1$ plate, we denote the dimension of effective perspective images segments as $l_{LCD}$. According to the geometric relationship, there is $l_1 = l_2 = l_{LCD} = l$. As shown in Figure 7, to determine the number of the hogel on

the virtual $H_1$ plate, we first determine the number of segments on the LCD panel, which is presented by $n_{LCD}$. Choosing the hogel on $H_1$ plate corresponds to the hogel on the $H_2$ plate, then

$$n_{LCD} = \begin{cases} \frac{L_{LCD}}{l}, & \text{if} \quad \frac{L_{LCD}}{l} \text{ is odd} \\ \frac{L_{LCD}}{l} - 1, & \text{if} \quad \frac{L_{LCD}}{l} \text{ is even} \end{cases} \tag{1}$$

where $n_{LCD}$ should be odd here, $n_{hogel}$ represents the segment number on both sides of the center segment on the LCD panel, $n_{hogel} = \frac{n_{LCD}-1}{2}$.

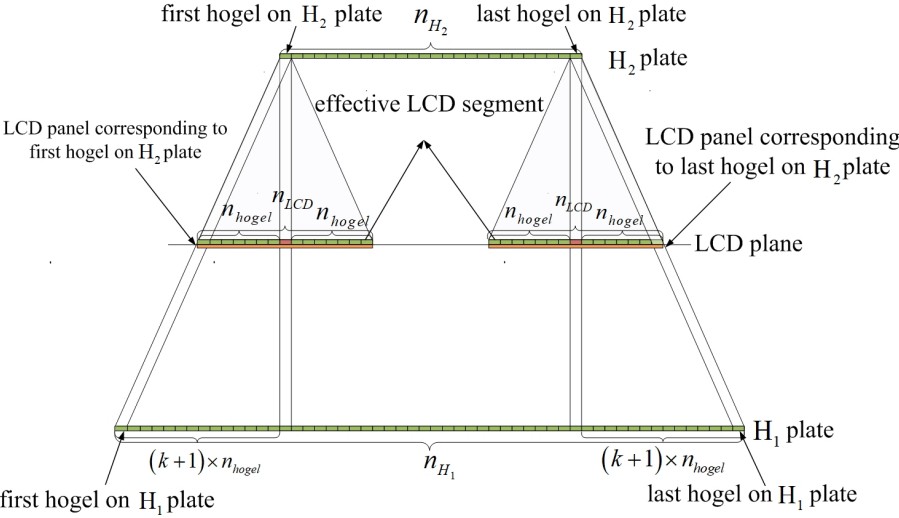

**Figure 7.** Determination of the hogel number for holographic element-based EPISM methods at $k = 1$.

Introducing an index $k$, $k$ represents the ratio of the distance from the LCD panel to virtual $H_1$ plate and $H_2$ plate, and $k = \frac{L_1}{L_2}$. According to the geometric relationship, when $k = 1$, the hogel corresponding to the adjacent segment of the LCD panel is separated by one hogel. When $L_1$ and $L_2$ are not equal, let $k$ be a positive integer; this means that the adjacent segment of the LCD panel has corresponding hogels on $H_1$ plate. The interval hogel number between adjacent hogels is exactly equal to $k$.

The number of hogels on the $H_1$ plate can be represented as $n_{H_1} = 2 \times n_{hogel} \times (k+1) + n_{H_2}$.

### 3.2. Effective Perspective Image Segmentation and Mosaicking

As shown in Figure 8, for the $i$th hogel on the $H_2$ plate, the order of the hogel on the virtual $H_1$ plate that corresponds to the $i$th hogel on the $H_2$ plate should be $i + n_{hogel} \times (k+1)$. For the $i$th hogel on the $H_2$ plate, the segment on the center of the LCD panel is denoted as the $0th$ segment, and let $m = \left(-n_{hogel}, n_{hogel}\right)$, the order of hogel on the virtual $H_1$ plate that corresponds to the $m$th segment of the LCD panel and $i$th hogel on the $H_2$ plate is $i + n_{hogel} \times (k+1) + m \times (k+1)$, and this is the hogel we are looking for. Left endpoint of the hogel on the virtual $H_1$ plate that corresponds to the $0th$ segment of the LCD panel is denoted as $x_0$; according to the previous results, we have $x_0 = \left[i + n_{hogel} \times (k+1) - 1\right] \times l_1$, left end the hogel on the $H_1$ plate that corresponds to the $m$th segment of LCD denoted as $s$, $s = \left[i + m \times (k+1) + n_{hogel} \times (k+1) - 1\right] \times l_1$, $h = (x_0 - s) \times \left(1 - \frac{1}{k+1}\right) \times l_1$ represents the distance between the corresponding effective perspective images segments and the location of hogel on $H_1$ plate. The left end coordinates of the segment are denoted as $e$,

$$e = \left[\left(h - \frac{1}{2}\right) \times l_1 + \frac{L_{LCD}}{2}\right] \times 100 + 1 \tag{2}$$

and the right end coordinates of the segment denote as $f$,

$$f = \left[\left(h - \frac{1}{2}\right) \times l_1 + \frac{L_{\text{LCD}}}{2}\right] \times 100 + e_p \tag{3}$$

where $e_p = l_1 \times 100$, according to the left and right ends of the segment, determine the $m$th effective perspective images segments. By mosaicking all the selected effective perspective images segments in sequence, we obtain the synthetic perspective image corresponding to the $i$th hogel on $H_2$ plate. By exposing the hogel on $H_2$ plate in sequence, the holographic stereogram is obtained.

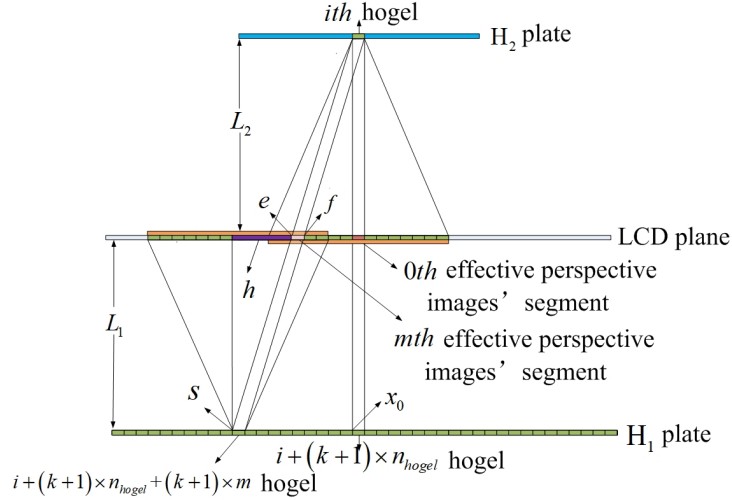

**Figure 8.** Diagram of effective perspective images segments.

## 4. Experimental Verification

An LCD panel was used for exposing holographic stereogram, for convenience in calculation, the corresponding resolution on 1 cm × 1 cm display area of LCD panel was $100 \times 100$ pixels. To this end, the Panasonic LCD panel (VVX09F035M20) had been selected, its size was 8.9 inches, and the resolution was $1920 \times 1200$. Select 10 cm × 10 cm area in the center of LCD panel as the effective display area. Thus, the resolution of the synthetic perspective image for exposure is $1000 \times 1000$ pixels.

A mapped teapot model is used as the 3D scene. The depth was 4.8 cm, the height was 3 cm, the width was 4.2 cm, and it was tipped 40°. Let $k = 1$, $l_1 = l_2 = 0.2$ cm, $L_1 = L_2 = 18.6$ cm. A camera was set in front of the teapot, and the FOV was 30°. The size of $H_2$ plate was 6 cm × 6 cm, $n_{H_2} = 30$, and the hogel number of $H_2$ plate was $30 \times 30 = 900$. According to the previous formula, $n_{H_1} = 126$, the number of the camera position should be $126 \times 126 = 15{,}876$.

As shown in Figure 9, the synthetic perspective images of the holographic element-based EPISM method are given, $\text{image}(6, 16)$ is the synthetic perspective image corresponding to the order number of the sixth row and the sixteenth column's hogel on $H_2$ plate. As EPISM methods, the synthetic perspective images are pseudoscopic images, and the reconstructed scene can reproduce the recording scene truthfully in correct depth.

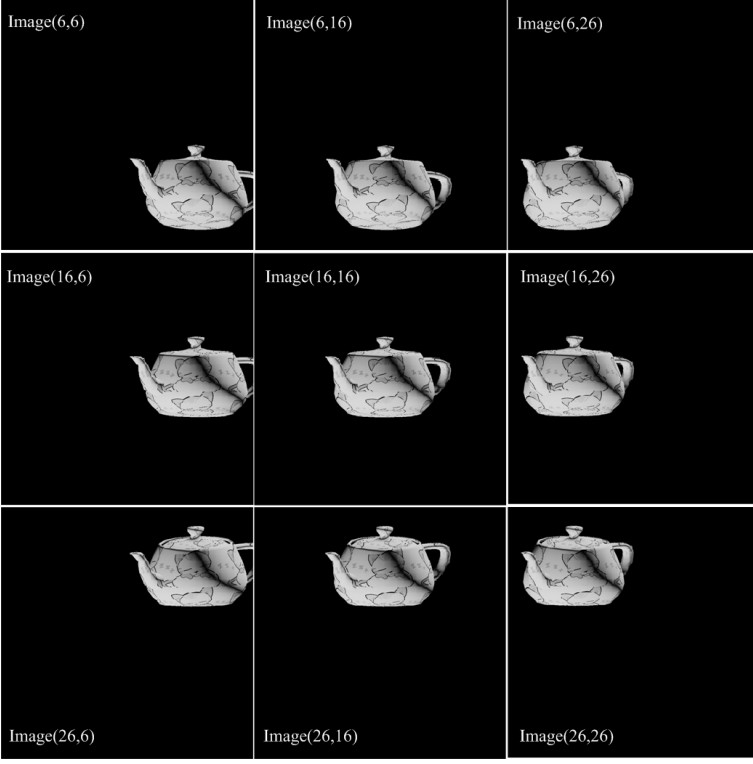

**Figure 9.** Synthetic perspective image of the holographic element-based EPISM methods.

As shown in Figure 10, the optical setup using for holographic-based EPISM method holographic stereogram is presented. A 639 nm custom-made red laser was used as the laser source for holographic stereogram printing. The max power of laser source was 1.2w. An electric shutter was used to control the exposure time. Its model was Sigma Koki SSH-C2B. The direction of laser beam was changed by a reflector. A non-polarizing beam splitter(NPBS) was set in the new direction of the laser beam, and the laser beam was divided into two laser beams perpendicular to each other. The laser beam vertical to the original direction was denoted as signal beam, and the laser beam in the same direction as the original direction was denoted as reference beam. An attenuator was set in the same direction of signal beam to adjust the intensities, and a spatial filter was used to expand the signal beam. The LCD panel introduced earlier was used to load the synthetic perspective image, a diffuser was used to scatter the signal beam to the hogel aperture, and the diffuser was placed close to the LCD panel. The reference beam direction was changed by a reflector, and an attenuator was used to adjust the intensities of the reference beam. Another reflector was used to change the reference beam to the backside of holographic plate, and a spatial filter was used to expand the reference beam. Then, the expanded beam was changed into a uniform plane wave by a collimating lens; the focal length of the lens is 75 mm. A manual ultrafine silver-halide plate for He-Ne laser was used as holographic plate; its grain size was about 10–12 nm. The holographic plate placed 18.6 cm away from LCD panel, two diaphragms with apertures are placed on both sides of the LCD panel, the signal beam and reference beam passing through the aperture interfere on the holographic plate to generate hogel. A X-Y motorized stage(KSA300, MC600) was used to carry the holographic plate; it can move both on the horizontal and vertical rail. The step of the X-Y motorized stage was $l_2$. A computer was used to time-synchronously control the shutter, the LCD panel, and the motorized stage.

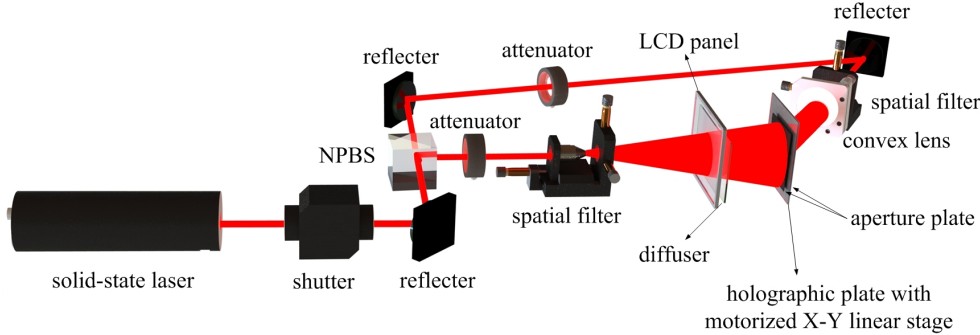

**Figure 10.** The optical setup of the synthetic holographic stereogram printing system.

The synthetic perspective image used to expose the hogel on the $H_2$ plate needs to be flipped horizontally, because the LCD panel corresponding to the virtual $H_1$ plate and the LCD panel corresponding to the $H_2$ plate were in the opposite direction.

Equation $T_e = E/(P_s + P_r)$ was used to express the exposure time, where $P_s$ is the intensity of signal beam energy, $P_r$ is the intensity of reference beam energy, and $E$ denotes as the light sensitivity of holographic plate. In this experiment, $E = 1250 \, \mu\text{J/cm}^2$, $P_s = 10 \, \mu\text{J/cm}^2$, $P_r = 300 \, \mu\text{J/cm}^2$. The energy ratio between $P_s$ and $P_r$ was selected as 1:30; this ratio can greatly reduce the printing time on the premise of guaranteeing the image quality, and the exposure time was $T_e = 4$ s. The waiting time was 4 times as much as exposure time to reduce the vibration result, and the waiting time $T_w = 16$ s. The printing time was the sum of exposure time and waiting time, the hogel's printing time $T_h = T_e + T_w = 20$ s, and total printing time $T_t = T_h \times N_{H_2} \times N_{H_2} = 18,000$ s.

Figure 11 shows the reconstruction images of an optical experiment result. A conjugate beam of reference beam was used to illuminate the holographic plate, and a camera used to record the reconstruction image of holographic plate. Its model is Canon EOS 5D-mark3 DS126091, the focal length of camera lens is 100 mm, the camera was set 50 cm away from holographic plate, and a real image of 3D scene was captured by camera.

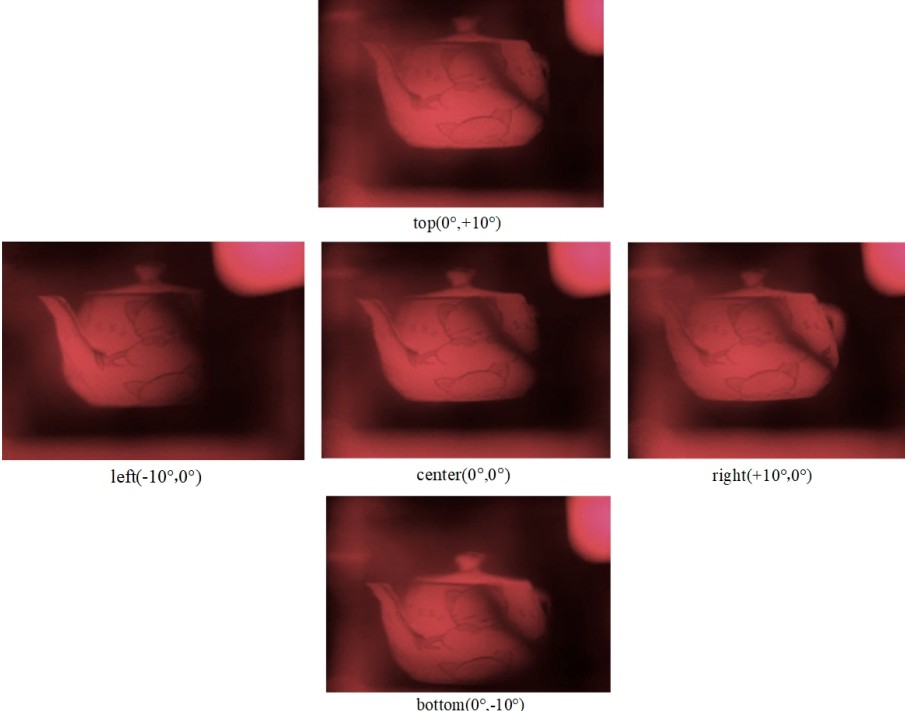

**Figure 11.** Optical reconstruction images in different viewing position obtained by experiments.

As shown in Figure 11, by the effective display size and the distance $L_2$, the FOV of $H_2$ plate is about 30°. The details of the original scene are perfectly reproduced, and full parallax information can be presented by the holographic element-based EPISM method. When the camera position is close to the limited FOV, due to the use of simple camera sampling, the reconstructed scene is as incomplete as the original scene, with the observation area unable to achieve 30° FOV.

To confirm the position of reconstructed image, we put two rulers together with the holographic plate. A camera was put 50 cm away from holographic plate, and the result is shown in Figure 12. The position relation of the rulers and holographic plate are shown in Figure 12a; the ruler on the left side is on the same plane as the holographic plate, and the ruler on the right side is on the position of the reconstruction image, away from the holographic plate by 18.6 cm. In Figure 12b, focusing on the left ruler, both the figure on the ruler and the hogel on holographic plate are clear, and the figure on the right ruler is blurred. In Figure 12c, focusing on the right ruler, the figure on the ruler and the map on the reconstruction image are clear, and the left ruler is blurred.

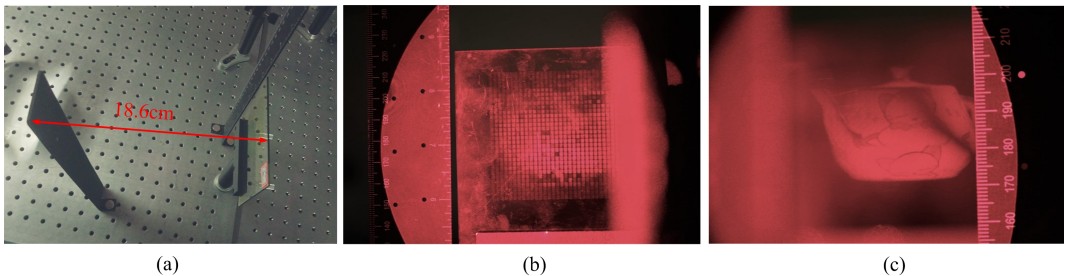

|     (a)     |     (b)     |     (c)     |

**Figure 12.** (**a**)The spatial position relation of two rulers and the holographic plate. (**b**) Focused on the left ruler, both rulers the holographic plate are clear. (**c**) Focused on the right ruler, both ruler and the surface of the teapot is clear.

As shown in Figure 13, the camera was also put in the position 50 cm away from the holographic plate; an EPISM method holographic stereogram was made according to the same configuration. Comparing the reconstructed images obtained by the EPISM method and the holographic element-based EPISM method, Figure 13a shows the position relation of rulers and the holographic plate; the farther ruler is placed in the position of the reconstruction image, with the ruler as far from the holographic plate as 18.6 cm, and the closer ruler is placed 1.5 cm behind the position of the farther ruler. Figure 13b,c show the reconstruction images of the EPISM method. Both the figure on the ruler and the map on the teapot are clear when focusing on the farther ruler in Figure 13b; as shown in Figure 13c, the figure on the ruler is clear and the map on the teapot is blurred when focusing on the closer ruler, and there is a mosaic misplacement on the map of the teapot. The reconstruction images of the holographic element-based EPISM method are shown in Figure 13d,e. As shown in Figure 13d, the figure on the ruler and the map of the teapot are clear when focusing on the farther ruler; the figure on the ruler is clear and the map on the teapot is only blurred when focusing on the closer ruler in Figure 13e.

Figure 14 shows the detail of Figure 13c,e, Figure 14a magnifies part of the Figure 13c, and the mosaic misplacement of the map of the teapot is revealed; Figure 14b magnifies the same part on the Figure 13e, and this part of the surface map of the teapot is blurred without mosaic misplacement. The brightness and contrast of the reconstructed image will be affected by the efficiency of the developer and the slight adjustment of the camera when shooting, but under the same conditions, the effective perspective images segments in EPISM method is smaller than in the proposed method, whether this difference will affect the quality of the reconstruction image still needs further study.

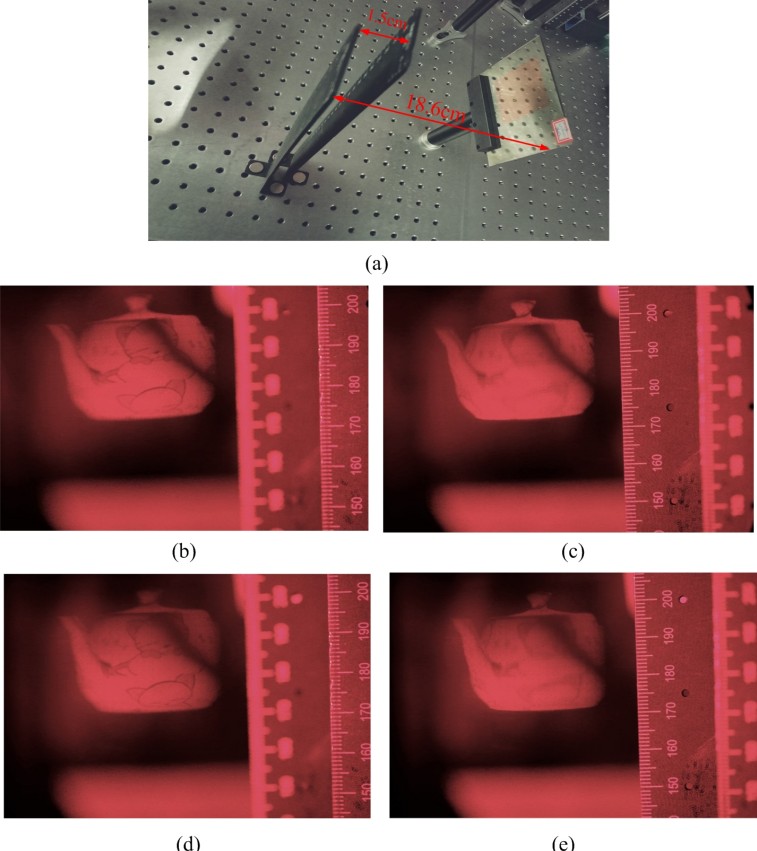

**Figure 13.** Comparison of reconstructed images between the EPISM method and the holographic element-based EPISM method. (**b**,**c**) are the reconstructed images of the EPISM method, (**d**,**e**) are the reconstruction images of the holographic element-based EPISM method. (**a**) The spatial position relation of two rulers and the holographic plate. (**b**) The reconstruction image of the EPISM method when focused on the farther ruler. (**c**) The reconstruction image of the EPISM method when focused on the closer ruler. (**d**) The reconstruction image of the holographic element-based EPISM method when focused on the farther ruler. (**e**) The reconstruction image of the holographic element-based EPISM method when focused on the closer ruler.

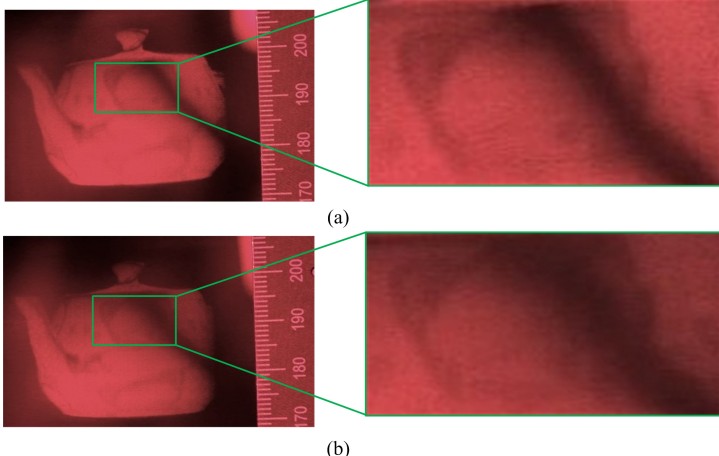

**Figure 14.** Comparison of details between reconstructed images of the EPISM method and reconstructed images of the holographic element-based EPISM method (**a**) Details of reconstructed images of the EPISM method (**b**) Details of reconstructed images of the holographic element-based EPISM method.

The experimental results show that the holographic element-based EPISM method can reproduce the 3D scene as well as the EPISM method, and solve the problem of mosaic misplacement in the EPISM method, but there are some restrictions on the EPISM method. The condition $l_1 = l_2$ and $L_1 = L_2$ must be satisfied so that there will be a hogel on virtual $H_1$ plate corresponding to the area formed by hogel correspondence between $H_2$ plate and LCD panel. In the future, we will consider how to generate effective perspective image segments when we relax this restriction appropriately.

## 5. Conclusions

In this paper, holographic element-based effective perspective image segmentation and mosaicking method was proposed for holographic stereogram printing. By using hogel correspondence to obtain the effective perspective images segments, the mosaic misplacement in the EPISM method is avoided, and high-quality reconstruction image also can be obtained as the EPISM method; however, in order to establish hogel correspondence, some restrictions are introduced in the proposed method. We will try to remove these restrictions in future work.

**Author Contributions:** Conceptualization, F.F. and X.J.; Methodology, F.F.; Software, J.W.; Validation, J.W.; Formal Analysis, S.C.; Writing—Original Draft Preparation, T.Z.; Writing—Review & Editing, C.H.; Supervision, X.Y.; Project Administration, X.J.; Funding Acquisition, X.Y.

**Funding:** This research was funded by [the National Key Research and Development Program of China] grant number [2017YFB 1104500], [National Natural Science Foundation of China] grant number [61775240], [Foundation for the Author of National Excellent Doctoral Dissertation of the People's Republic of China] (FANEDD) grant number [201432].

**Conflicts of Interest:** The author declares no conflict of interest.

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
