# Peer review of "Holographic Element-Based Effective Perspective Image Segmentation and Mosaicking Holographic Stereogram Printing"

_applsci, doi:10.3390/app9050920_

Round 1
Reviewer 1 Report
In this article the authors propose a new technique which the call “holographic element based effective perspective images’ segmentation and mosaicking”. This technique means an improvement over the “Effective perspective images’ segmentation and mosaicking” (EPISM) technique. The main difference of the two methods is the use of holographic element (hogel) correspondence’ instead of ‘pixel correspondence’ to generate effective perspective images. The article is technically correct and well written. The method is well explained and supported by experimental evidence. I suggest to accept the article for publication. I only provide a few suggestions:
1) I suggest to revise the English
2) Line 58: in the article: “Figure.1” I suggest: “Figure 1”. This operation should be done for every call to a figure.
3) Line 104: “mosaicing”, it must say: “mosaicking”
4) Variables and quantity symbols should be in italic type. Unit symbols should be in roman type. The numbers and units must be separated by a space. For instance Line 163: the authors: “l1 = 0.2cm”, the correct expression: “l1 = 0.2 cm”
5) Line 136: k must be a positive integer (not an integer only).
Author Response
Ø 1) I suggest to revise the English |
Response: Thank you very much for your sincere advice. The English of the full text have been checked and revised carefully as suggested.
Ø 2) Line 58: in the article: “Figure.1” I suggest: “Figure 1”. This operation should be done for every call to a figure. |
Response: Thank you very much for your sincere advice. “Figure.” is replaced by “Figure ” for every call to a figure in this paper.
Ø 3) Line 104: “mosaicing”, it must say: “mosaicking” |
Response: Thank you very much for your sincere advice. We are so sorry for such mistakes, and the “mosaicing” is replaced by “mosaicking” in this paper.(Line 92, 103,152)
Ø 4) Variables and quantity symbols should be in italic type. Unit symbols should be in roman type. The numbers and units must be separated by a space. For instance Line 163: the authors: “l1 = 0.2cm”, the correct expression: “l1 = 0.2 cm” |
Response: Thank you very much for your sincere advice. Symbols have been corrected, and the numbers and units are separated in this paper.
Ø 5) Line 136: k must be a positive integer (not an integer only). |
Response: Thank you very much for your sincere advice. As you point out, k should be a positive integer, and we only consider the case that k should be an integer, and ignoring that our hypothesis is not valid when k is a negative integer, this problem has been corrected in paper.(Line 135)

Reviewer 2 Report
Dear Authors,
I would suggest your manuscript for the publication when the following changes is introduced.
In general: Punctuation should be carefully checked in the paper. In many places comma is used instead of full stop. Also, English gramma should be carefully checked.
Abstract
Line 3-8. I would suggest removing all ‘and divide the sentence as follows.
In this paper a method named as holographic element based effective perspective images’ segmentation and mosaicking wais proposed,-. a ‘hHolographic element (hogel) correspondence’ is used in EPISM methodin the same wayas ‘pixel correspondence’ is used in direct-writing digital holography (DWDH) method to generate effective perspective images segments, .and tThe synthetic perspective image for holographic stereogram printing is obtained by mosaicking all the effective perspective images segments.
Main body
Line 13. The sentence “Holographic stereogram printing technology is used in many fields such as medical, exhibition, military, art and so on.” should be rewritten. Please find another words to describe the areas of application. For example, “Medicine” instead of “medical”. Also, I would suggest adding references to support the statement.
Line 22. Bad English gramma in the sentence: “Then the second-step process of recording the reproduced image of the master holographic plate onto the transfer holographic plate.” Please correct it.
Line 33-35. Gramma mistakes: “This technique employs a pixel corresponding method, replaces the image pixels loaded onto the SLM with the pixels of the perspective image, and accurately reproduces the recorded scene.
Fig.1-8, 10. I would suggest increasing the font size of description on the figures.
Fig. 6. I would suggest to enlarge the figure.
Line 66-68. Could authors add the minimum number of hogel required to obtain high quality holographic stereogram in the following sentence. “The ‘pixel correspondence’ of the DWDH method can accurately reproduce the 3D scene, but the resolution of the sampled image is determined by the number of hogels, high quality holographic stereogram cannot be obtained when the number of hogels is small.”
Title of Fig.7. Typo - “hoegel”
Line 185. Please include the type/material/producer of the holographic plate used. What is the absorbance band of the holographic plate?
Line 198. Typos –upper case missed- Pr = 300µJ/cm2
Line 248. Authors claimed “some restrictions are introduced in the proposed method; we will try to remove these restrictions in our future study.” These restrictions must be properly described in the paper as well as possible ways to overcome the restrictions.To make a final conclusion if this method provides any benefits, it is crucial to know all the restrictions arised.
In my opinion, the contrast and brightness of the image obtained by the holographic element based EPISM method is worse than the one obtained by the EPISM method. How authors can explain this?

Author Response
Ø 1) In general Ø Punctuation should be carefully checked in the paper. In many places comma is used instead of full stop. Also, English gramma should be carefully checked. |
Response: Thank you very much for your sincere advice. The punctuation and grammar of the full text have been carefully checked and revised, and many minor problems have been corrected.
Ø 2) Abstract Line 3-8. I would suggest removing all ‘and divide the sentence as follows:….. |
Response: Thank you very much for your sincere advice, and the abstract was rewritten as recommended: Effective perspective images’ segmentation and mosaicking (EPISM) method is an effective holographic stereogram printing method, but a mosaic misplacement of reconstruction image occurred when focusing away from the reconstruction image plane, In this paper a method named as holographic element based effective perspective images’ segmentation and mosaicking is proposed. Holographic element (hogel) correspondence is used in EPISM method as pixel correspondence is used in direct-writing digital holography (DWDH) method to generate effective perspective images segments. The synthetic perspective image for holographic stereogram printing is obtained by mosaicking all the effective perspective images segments. Optical experiments verified that the holographic stereogram printed by the proposed method can provide high-quality reconstruction image and solve the mosaic misplacement inherent in the EPISM method.
Ø 3)Main body Ø 3.1) Line 13. The sentence “Holographic stereogram printing technology is used in many fields such as medical, exhibition, military, art and so on.” should be rewritten. Please find another words to describe the areas of application. For example, “Medicine” instead of “medical”. Also, I would suggest adding references to support the statement. Ø |
Response: Thank you very much for your sincere advice. In order to avoid confusion between the application of holographic stereogram and the application of hologram in some specific fields, the sentence was changed into general description as follow: Holographic stereogram printing technology is used in many fields.(Line 13)
Ø 3.2) Line 22. Bad English gramma in the sentence: “Then the second-step process of recording the reproduced image of the master holographic plate onto the transfer holographic plate.” Please correct it. |
Response: Thank you very much for your sincere advice. We have changed the sentence to be the following: “Then the second-step of the process is to record the reproduced image of the master holographic plate onto the transfer holographic plate.”(Line 21)
Ø 3.3) Line 33-35. Gramma mistakes: “This technique employs a pixel corresponding method, replaces the image pixels loaded onto the SLM with the pixels of the perspective image, and accurately reproduces the recorded scene. |
Response: Thank you very much for your sincere advice. We are so sorry for such mistakes.
Gramma mistakes are corrected as follow: This technique employs a pixel corresponding method, replaces the image pixels loaded onto the SLM with the pixels of the perspective image, and accurately reproduces the recorded scene. (Line 32-34)
Ø 3.4) Fig.1-8, 10. I would suggest increasing the font size of description on the figures. |
Response: Thank you very much for your sincere advice. The font size of description on
Fig 1-8, 10. were increased as suggest.
Ø 3.5) Fig. 6. I would suggest to enlarge the figure. |
Response: Thank you very much for your sincere advice. The Fig 6. was enlarged as suggested.
Ø 3.6) Line 66-68. Could authors add the minimum number of hogel required to obtain high quality holographic stereogram in the following sentence. “The ‘pixel correspondence’ of the DWDH method can accurately reproduce the 3D scene, but the resolution of the sampled image is determined by the number of hogels, high quality holographic stereogram cannot be obtained when the number of hogels is small.” |
Response: Thank you very much for your sincere advice. The minimum number of hogel required to obtain high quality holographic stereogram was added as follow: The pixel correspondence of the DWDH method can accurately reproduce the 3D scene, but the resolution of the sampled image is determined by the number of hogels, high quality holographic stereogram often require hundreds of thousands of sampled images.(Line 65-67)
Ø 3.7) Title of Fig.7. Typo - “hoegel” |
Response: Thank you very much for your sincere advice. “hoegel” was corrected into “hogel”.(Title of Fig 7)
Ø 3.8) Line 185. Please include the type/material/producer of the holographic plate used. What is the absorbance band of the holographic plate? |
Response: Thank you very much for your sincere advice. The holographic plate was a manual ultrafine silver-halide plate, and the absorbance band of holographic plate was for He-Ne laser, its grain size was about 10-12 nm. The information of the holographic plate is added to the paper as suggest.(Line 186-187)
Ø 3.9)Line 198. Typos –upper case missed- Pr = 300µJ/cm^2 |
Response: Thank you very much for your sincere advice. The mistake was corrected in the paper as suggest.(Line 198)
Ø 3.10)Line 248. Authors claimed “some restrictions are introduced in the proposed method; we will try to remove these restrictions in our future study.” These restrictions must be properly described in the paper as well as possible ways to overcome the restrictions.To make a final conclusion if this method provides any benefits, it is crucial to know all the restrictions arised. |
Response: Thank you very much for your sincere advice. Restrictions of proposed method were described in the paper and the description had been changed as follow: but there are some restrictions than the EPISM method, The condition l1=l2 and L1=L2 must be satisfied so that there will be a hogel on virtual H1plate corresponding to the area formed by hogel correspondence between H2 plate and LCD panel. In the future, we will consider how to generate effective perspective images segments when we relax this restriction appropriately. (Line 245-248)
Ø 3.11)In my opinion, the contrast and brightness of the image obtained by the holographic element based EPISM method is worse than the one obtained by the EPISM method. How authors can explain this? |
Response: Thank you very much for your sincere advice. The brightness and contrast of the reconstructed image will be affected by the efficiency of the developer and the slight adjustment of the camera when shooting, but under the same conditions, the effective perspective images segments in EPISM method is smaller than in proposed method, whether this difference will affect the quality of the reconstruction image still needs further study.(Line 238-242)